# Perception of People Living with HIV and Healthcare Workers on Differentiated Service Delivery Programs in Nigeria: A Qualitative Study

Johnbaptist Ezenduka[1], Peter Nwaokennaya[2], Gbenga Benjamin Obasa[1], Geoffrey Ogbeke[2], Ogonna Onuorah[1], Lawal Abubakar[1], Adebobola Bashorun[2], Ginigeme Ogochukwu[1], Emerenini Franklin[1]*, Adewale Akinjeji [1,3]

**1** ICAP Nigeria, Abuja, Nigeria, **2** National AIDS and STI Control Program, Ministry of Health, Abuja, Nigeria, **3** Department of Global Public Health, Karolinska Institutet, Stockholm, Sweden

* fe2204@cumc.columbia.edu

## Abstract

Nigeria, home to approximately 1.9 million people living with HIV (PLHIV), faces significant challenges in providing adequate care and treatment to the teeming number of PLHIV, particularly following the adoption of the 'test and treat' policy and the HIV 'SURGE' initiative. The Differentiated Service Delivery (DSD) model was introduced to address the increased burden on the healthcare system, offering patient-centered care through diverse methods like community-based care, fast-track drug refills, and task-shifting to lower-level healthcare providers. This study, conducted by the National AIDS and STI Control Program (NASCP) in collaboration with ICAP at Columbia University, evaluates the perspectives of healthcare workers and PLHIV on DSD programs in Nigeria.

This is a qualitative study involving key informant interviews (KIIs) with 12 healthcare workers and focus group discussions (FGDs) with 153 PLHIV across four Nigerian states (Anambra, Kaduna, Lagos, Taraba). Healthcare workers with at least one year of ART service experience and PLHIV on ART for at least one year with suppressed viral loads were included in the study. Data was analyzed using an inductive, thematic approach to identify emergent patterns and themes. The study revealed that DSD models, such as peer-led Facility ART groups and community pharmacy ART refills, were widely implemented. The primary factors influencing the deployment of DSD models included client convenience and the need to alleviate healthcare worker load. Positive experiences with DSD were associated with convenience, confidentiality, and affordability, whereas negative perceptions stemmed from high service costs, poor healthcare worker attitudes, and confidentiality concerns. Additionally, the study highlighted the necessity for continuous training of healthcare workers, engagement with PLHIV, and increased awareness programs to improve DSD service delivery.

**Data availability statement:** "Most relevant data is within the manuscript and its Supporting Information files. Qualitative data in the form of interview transcripts has not been made publicly available in order to protect vulnerable study participants. However, qualitative data can be requested from this manuscript's Corresponding Author via email at adewale. akinjeji@ki.se"

**Funding:** The author(s) received no specific funding for this work.

**Competing interests:** The authors have declared that no competing interests exist.

Overall, the study underscores the importance of addressing financial barriers, enhancing healthcare worker training, and ensuring confidentiality in optimizing DSD models and improving HIV care outcomes in Nigeria.

## Introduction

Nigeria is among the countries that have the highest burden of HIV globally [1]. With an estimated 1.9 million people living with HIV (PLHIV) [2], effective management and treatment remain critical priorities for public health initiatives in the country. In 2016, Nigeria adopted the 'test and treat' which stipulates a rapid ART initiation for all persons who test HIV positive, and the subsequent implementation of the HIV 'SURGE' resulted in a rapid scale-up of HIV case finding and increased number of Persons Living with HIV placed on treatment thereby placing a higher-than-usual burden on the fragile healthcare system [3–6]. Differentiated Service Delivery (DSD) model of care was among the innovative approaches adopted to address the challenges associated with a high volume of persons in HIV care and treatment in a country with scarce resources [7].

Differentiated Service Delivery refers to a patient-centered approach to HIV care, aiming to tailor services to the diverse needs of PLHIV [8]. These include various models, such as community-based care, fast-track drug refills, and task-shifting responsibilities to lower-level healthcare providers [9]. DSD programs seek to enhance treatment adherence, Continuity of treatment, and overall health outcomes for PLHIV while alleviating the strain on healthcare systems by offering flexible service delivery options [10,11]. Nigeria officially adopted the DSD approach in line with WHO recommendations in 2019 and has since scaled the implementation across the 36+1 subnational units of the federation [12]. The rapid scale-up of DSD contributed to the resilience of the HIV program in-country during the COVID-19 pandemic [12]. The COVID-19 pandemic and the resulting global response reinforced the need for countries to adopt and evolve sustainable mechanisms for managing chronic conditions, including access to ART for PLHIV. Nigeria launched the DSD Operational manual in 2021 [13] this document provides the framework for the implementation of DSD in Nigeria. The document recognizes ten different DSD models subdivided into facility-based models and community-based models. The different models of differentiated ART service delivery are summarized in Table 1 below.

Despite the potential benefits of DSD programs which has been documented to include reducing cost of accessing care, increasing convenience for recipients of care, [14,15], and comparable outcomes with regards to retention and viral suppression among care recipients [16–18], there is a need to regularly review the effectiveness of services and activities in these settings. A critical aspect of evaluating the effectiveness of DSD involves assessing the perceptions and experiences of both PLHIV and healthcare professionals (HCWs) PLHIV. Understanding their perspectives is crucial for identifying barriers to access, adherence, and successful program implementation [19]. Therefore, exploring the perceptions of PLHIV and HCWs

**Table 1. Models of differentiated ART service Delivery in Nigeria.**

| Facility-based | Community-based |
|---|---|
| Fast track (Individual or Group): stable clients pick their drugs from the facility pharmacy without going through the normal clinic flow, including a doctor's review | Community drug distribution points: These are designated points within the community where ARVs and other medications are dispensed to stable PLHIV. |
| Health Facility Based ART Group: These are health facility-based groups formed voluntarily by support groups of PLHIV who are already meeting regularly at the health facility for ARVs and other medication refills. This can either be client-led or healthcare provider-led. | Community Pharmacy Refills: This is the pickup of drugs by PLHIV from designated or selected community pharmacies. |
| Multi-month dispensing**: Medication dispensing interval of 3 months and above. *This is not considered a stand-alone DSD model but rather as a refill mechanism that applies to all the different DSD models.* | Community ART Group (CAG): These are community-based groups formed voluntarily by PLHIV within a community for ARVs and other medication refills. This can either be PLHIV led, or healthcare worker led. These healthcare workers may include community health workers, |
| Decentralization: refers to the devolution of stable clients from larger, centralized secondary and tertiary facilities (hubs) to smaller more peripheral primary facilities (spokes). This can be: - semi-autonomous model restricts ART service delivery at PHCs to ARV/medication refills - autonomous model allows for ART initiation at the PHC level, ARV, and medication refills. | |
| Adolescent clubs: Groups of adolescents and young people living with HIV for whom age-appropriate, affordable, friendly health services are provided in an accessible and acceptable environment | |
| Post Natal Clubs: Groups of women living with HIV who are supported in the postnatal period by healthcare workers and other volunteers like mentor mothers to ensure improved maternal/child health outcomes | |
| One-Stop Shops and Mobile Clinics are community-based service delivery sites where multiple services are offered, and clients can access all their needs under one roof targeted specifically at providing services for Key Populations. | |
| Peer-led: These groups are facilitated by trained peer educators. These educators are trained in screening for HIV, opportunistic infections, and self-testing to offer community testing. | |

regarding DSD programs in Nigeria is essential for informing policy decisions, improving program design, and ultimately advancing the quality of HIV care and treatment services in the country.

The National AIDS and STI Control Program (NASCP) of the Nigeria Federal Ministry of Health collaborated with ICAP at Columbia University to assess the perspectives of healthcare professionals and PLHIV on Differentiated Service Delivery programs in Nigeria to gain comprehensive insights into the efficacy and impact of these programs on HIV care and treatment in the country.

## Methodology

### Setting and recruitment

This qualitative study, part of a larger mixed-methods investigation, aimed to understand contextual factors influencing differentiated service delivery (DSD) models' uptake and retention in four socio-culturally diverse Nigerian states (Anambra, Kaduna, Lagos, Taraba). The states were conveniently selected to ensure sociocultural and geographic representation of the country as part of the DSD program performance review conducted under the NASCP GC6 grant. While a few Focus Group Discussion (FGD) participants are rural residents, all the Key Informant Interview (KII) participants are healthcare workers providing HIV and other health services in health facilities located at urban and peri-urban locations. The healthcare workers recruited for the KII are persons who received some formal training, e.g., Doctors, Nurses, CHEWs etc., and have received either a formal or informal training on HIV service provision for PLHIV. Participant experiences were captured through key informant interviews with healthcare workers and focus group discussions with people living with HIV.

Qualitative data were collected through semi-structured interviews and discussions, employing audio recording, transcription, and confidentiality protocols.

**Key informant interviews.** Twelve key informant interviews were conducted with healthcare workers providing HIV services in Anambra, Kaduna, Lagos, and Taraba states (3 per state). Healthcare providers with more than one year

of experience providing ART services were purposively selected as Key informants. KII participants were purposively selected to ensure representation of different facility types including KP One-Stop Shops (OSS), Primary, Secondary, and Tertiary Health facilities, Trained interviewers used a semi-structured guide to facilitate in-depth discussions on differentiated service delivery implementation and service provision. The face-to-face interviews, conducted in English, aimed to elicit comprehensive insights from frontline workers operationalizing DSD models, allowing exploration of predefined and emerging topics while ensuring consistency through the interview guide and interviewer training. The KII guide explored objective parameters such as the type of DSD model available in the facility and the models preferred by recipients while subjectively assessing why recipients of care either accepted or declined devolvement. Participant responses were audio-recorded and transcribed verbatim before analysis.

**Focus group discussions.** Sixteen FGDs involving 153 participants were conducted, stratified into key populations (4), general adult population stratified by gender (8), and young adults aged 18–24 years (4). Participants were stratified in this manner to enable a deeper exploration of the DSD experience by the different categories of recipients of care. All the participants had been on ART for ≥12 months with suppressed viral loads. FGDs explored experiences, challenges, and recommendations related to DSD models. The discussions were conducted in local Nigerian languages, Igbo, Hausa, Yoruba, and Pidgin, by two interviewers fluent in the local languages across each state. Audio recordings of the interviews were transcribed and translated by certified translators before analysis. Recruitment for KIIs and FGDs followed standardized procedures, with eligibility screening and assurances of voluntary participation without adverse impacts. Participants from the key population were selected to ensure representation of different KP typologies, including Sex Workers (SW), Persons Who Inject Drugs (PWID), Men who have Sex with Men (MSM), and Transgender (TG) persons.

## Inclusion and exclusion criteria

### Inclusion criteria.

**Key informant interviews** Key informants who are adults (Aged 18 years) have provided HIV services for at least one year and consent to participate.

**Focus group discussions** PLHIV over the age of 18 years who received care in the selected facilities and expressed their willingness to participate were considered eligible for inclusion. Also, participants must have been on ART for at least 12 months before the interview date and had achieved a suppressed viral load at their last assessment.

**Exclusion criteria.** Healthcare workers and PLHIV who did not consent to participate in the study were excluded from the qualitative analysis. To ensure the data's integrity and reliability, individuals who failed to respond to survey questions during the data collection process were also excluded from the study.

## Data analysis

The analysis followed an inductive, thematic approach, identifying emergent patterns and themes. Peer analysis, consensus discussions, and iterative coding of responses ensured rigor and reliability. The quantitative analysis involved descriptive statistics and thematic coding using Dedoose software, with a priori and emergent codes based on study objectives. Data were organized hierarchically to uncover related explanations and patterns.

## Ethical consideration and informed consent

Before data collection commenced, ethical approval was obtained from the Institutional Review Boards in Nigeria (NHREC/01/01/2007). Written Informed consent was obtained from all participants following a candid description of the objectives of the study, ensuring confidentiality and voluntary participation. Participants were informed about the study's purpose and rights, and only those who provided consent were included.

## Results/findings

### Key informant interview (KII) analysis

A total of 12 respondents participated in the KII, with an equal distribution of gender, comprising 50% male and 50% female as seen in Table 2. Among them, 33% were physicians, 25% were nurses, 17% were case managers and pharmacists, and 8% were public health specialists. Regarding experience providing ART services, 42% have less than five years, 33% between 6–10 years, and 25% more than ten years.

### DSD coverage

The availability of DSD models in facilities varied. The most common models across most facilities were the Support group -led Facility ART group, fast track, and mother-infant pair/mentor mother-infant pair. They were followed by. Community pharmacy ART refill. The other models varied in their availability across sites, as shown in Table 3 below.

### Deployment of DSD models

Several factors influenced the decision of healthcare workers and facilities to deploy different DSD models, and these factors are detailed in Table 4. The results show that client convenience and the need to reduce the healthcare worker load significantly influenced the model deployed in the facilities. However, the client's eligibility for a specific model and the need to improve patient care were of the slightest significance. Other factors considered included distance to facilities, cost of transportation, and the need to address stigma.

### Decision to deploy

Reasons for devolution of HIV/AIDS care services include prioritizing client convenience (50%), alleviating facility workload (50%), reducing long proximity to facilities (25%), tailoring care to fit client needs and clinical staging (16.67%), addressing transportation costs (16.67%), and minimizing stigmatization (16.67%).

Healthcare workers identified some of the key factors that informed their decisions to deploy specific DSD models. the need to prioritize client convenience and alleviate healthcare worker workload featured prominently in their responses. As one of the participants put it when asked what informed their decision to deploy their selected model, "*so that we can have quality care for the patient and to reduce patient waiting time. So I think it's for quality care, yes, and then to also enable us to handle the number of clients we have.*"

**Table 2. Demographics Characteristics.**

| Gender | N=12 |
|---|---|
| Male | 6(50%) |
| Female | 6(50%) |
| Profession | |
| Physicians | 4(33%) |
| Nurses | 3(25%) |
| Case Managers | 2(17%) |
| Pharmacists | 2(17%) |
| Public Health Specialists | 1(8%) |
| Years of Experience | |
| Less than 5 years | 5(42%) |
| 6 to 10 years | 4(33%) |
| More than 10 years | 3(25%) |

**Table 3. DSD Coverage.**

| DSD Coverage | N(%) |
|---|---|
| Facility ART group "Support group-led" | 11(92%) |
| Fast track | 10(83%) |
| Mother-infant pair/mentor mother-infant pair | 9(75%) |
| Community pharmacy ART refill | 8(67%) |
| Facility child/teen/adolescent club | 7(58%) |
| Home delivery | 7(58%) |
| Weekend and public holidays | 7(58%) |
| Community ART refill group "HCW-led" | 6(50%) |
| Community ART refill group "PLHIV-led" | 6(50%) |
| Facility ART group "HCW-led" | 5(42%) |
| After Hours | 5(42%) |
| One-Stop Shop (OSS) | 5(42%) |
| Adolescent community ART/Peer-led groups | 4(33%) |
| Decentralized (hub and spoke) | 2(17%) |

*This table represents a multiple response question. The results have been arranged to show the frequency of responses by each participant. The number of responses to each variable was divided by 12 (the total number of respondents) to get the percentage for each variable.

**Table 4. DSD Deployment.**

| Deployment of DSD Models | N=12 (100%) |
|---|---|
| Client convenience | 6(50%) |
| Alleviating facility workload | 6(50%) |
| Long proximity to facilities | 3(25%) |
| Best fit for the clients | 2(17%) |
| Clients Eligibility | 1(8%) |
| Client Schedule | 2(17%) |
| Tailoring care to client needs and clinical staging | 2(17%) |
| Addressing transportation costs | 2(17%) |
| Transportation | 2(17%) |
| Minimizing stigmatization | 2(17%) |
| Improve Patients Care | 1(8%) |

*This table represents a multiple response question. The results have been arranged to show the frequency of responses by each participant. The number of responses to each variable was divided by 12 (the total number of respondents) to get the percentage for each variable

Other factors such as the need to address transport cost featured less prominently, this is captured by this participants response.

"*because of time, most of them don't have time to come at the right appointment time. So people that don't want to come to the facility because they're scared of meeting someone they like, we can send them to the community. The one they love most is the home delivery. Because there is no out of pocket cost for them, the love home deliveries*"

**Reasons for declining devolvement**

Healthcare workers provided some reasons clients have either declined devolvement or voluntarily opted to be returned to a routine standard of care following devolvement to a DSD model. The most common reasons for this included the cost of services at the DSD service delivery points and confidentiality concerns. Other factors include poor healthcare worker attitude, a preference for facility services, proximity, and stigmatization. This is expressed by the participants response to reasons why recipients of care in their facility have either declined devolvement to a model, or freely opted to return to the facility after they have been devolved. One of the participants responded that "*What they say is that they are used to coming here, so they don't want to go to anywhere. Some of them are more comfortable when they are seen by a doctor. They tell you that they don't want a situation whereby you just give them medication and that is all because they may have some complaints that may not be big enough to take them to the hospital.*"

**Focus group discussions**

The Focus Group Discussions (FGDs) convened a total of 153 participants, consisting of 79 males (51.65%) and 74 females (48.35%), providing a balanced gender representation. Respondent characteristics presented in Table 5 show that most respondents are adults (above 18 years) who have been on ART for a minimum of 1 year and are familiar with DSD models. This is in line with the recommendations in the Nigeria guidelines for DSD eligibility [20]

**Themes, codes, and exemplar quotes**

The responses from the PLHIV were categorized into two main themes: barriers and enablers to the uptake of DSD services. These responses were then further divided into specific sub-themes presented below.

*Theme 1: Enablers.* The participants discussed factors contributing to a positive experience at DSD service delivery points, including convenience, confidentiality, and reduced service access costs. These factors promoted their uptake of DSD models and varied in their level of importance to the participants.

**Convenience** This was most important in shaping positive experiences for men and young adult participants. Factors contributing to increased convenience included proximity to pick-up points, flexible appointment scheduling, and streamlined processes. These factors significantly improved the participants' access to and utilization of HIV services. As one of the men puts it, "*I am currently enjoying the DSD program because they have been delivering my medications to my home for a while now. Due to my busy schedule, I often cannot make it to the facility in person, but I still go for my viral load tests.*" Another female participant has this to say "*I have benefitted from community pharmacy during the lockdown period. I was able to access my drugs and also do my viral load test in my area*". Another participant explained how DSD benefited them during the COVID19 lockdown. They said "*I have benefitted from community pharmacy during the lockdown period. I was able to access my drugs and also do my viral load test in my area. What I liked about the pharmacy was that during the lockdown, I was able to access the facility because it was close to me and it saved me transport and time*".

**Confidentiality** Confidentiality emerged as a key theme for the key population and women participants. This underscores the critical role of privacy in healthcare settings, which can foster trust and encourage individuals to seek and adhere to HIV-related care and treatment schedules. The confidential setting in some DSD service delivery points encouraged the uptake of services and retention in care. A member of the key population said, "*Home delivery is a better option because it prevents you from having to visit a community center to collect your medication. This way, you can avoid the possibility of running into someone you know. So, to avoid any potential embarrassment, home deliveries are a better option.*"

Another participant opined that "*Going to the pharmacy to pick up drugs is better. You know most people don't like it when people see them at the hospital, but at the pharmacy, you just walk up to the counter, and others around you assume it's a routine visit, not knowing it's for your medication. Also, there's no waiting compared to going to the hospital*"

**Table 5. FGD Demographics.**

| Responses | N= 153(%) |
|---|---|
| Gender | |
| Males | 79(51.65%) |
| Females | 74(48.35%) |
| State | |
| Lagos | 40(26.14%) |
| Taraba | 39(25.49%) |
| Anambra | 38(24.84%) |
| Kaduna | 36(23.53%) |
| FGD subgroup | |
| Men | 40(26.14%) |
| Key Population | 39(25.49%) |
| Women | 38(24.84%) |
| Adolescents | 36(23.53%) |
| Age group | |
| 18-24 years | 33(21.57%) |
| 25-39 years | 39(25.49%) |
| 40-49 years | 45(29.41%) |
| 50-64 years | 31(20.26%) |
| 65 and Above | 2(1.31%) |
| Decline | 3(1.96%) |
| Educational Status | |
| Completed Post- Graduate Degree | 7(5.00%) |
| Completed University | 28(18.00%) |
| Completed College of Education/Polytechnic | 45(29.41%) |
| Completed Secondary School | 63(41.18%) |
| Completed Primary School | 6(4.00%) |
| Never been to School | 1(0.41%) |
| Declined | 3(2.00%) |
| Number of years on ART | |
| Between 1 and 5 Years | 38(25%) |
| Between 6 and 10 Years | 44(30%) |
| Above 10 years | 65(42.5%) |
| Declined to answer | 5(3.%) |
| Have you heard about DSD? | |
| Yes | 120(79.00%) |
| No | 32(21.00%) |

**Affordability** Service affordability also emerged as another key theme that shaped the positive perception of care, especially among women. Affordability, in this case, refers to the cost of paying for the services and the cost of transport to the service delivery point. This suggests that removing financial barriers to healthcare access can significantly improve individuals' ability to seek and engage in HIV care. Initiatives to alleviate these costs are therefore highly valued by service users. A female respondent said, *"I receive home delivery, which helps reduce my transportation and the associated stress."* Another female also had this to say regarding saving transportation cost *"I like home delivery as it helps save on transportation. However, I dislike going to the hospital because of the time wasted there."* One of the young participants

reinforced the cost-saving enjoyed on some of the models. They said *"I also like the community pharmacy because it is good for patients that may not have transport fare to visit the clinic, so the person can walk to near pharmacy or take a car drop there."*

*Theme 2: Barriers.* Exploring the factors that can negatively influence the perception of DSD services among PLHIV showed varied responses from the participants. The key factors influencing the negative perception of PLHIV ranged from the high cost of services to confidentiality issues and the poor attitude of healthcare workers. These unpleasant experiences were experienced at the facility and community DSD models and negatively influenced participants' willingness to accept DSD and be retained in DSD service points.

**Cost of services**   The cost of services was significantly expressed as contributing to the negative perception experienced by adult male and female respondents. They highlighted the financial burden associated with accessing HIV services, including testing, treatment, and medications in some DSD service delivery points and health facilities. They expressed that they must pay between 1,000 (1 USD) and 5,000 naira (5 USD) before receiving services, including fast-track in some instances. These high costs may act as barriers to accessing care in their preferred models and negatively influence their willingness and experiences with these services. As one of the respondents put it

*"The fees at facility XX have been a problem, especially since I travel from my town with my daughter to collect my medication there. Because of the charges, I requested a transfer to another facility for treatment."*

The fees at the facility they mentioned have been a problem, especially since I travel from Umunze with my daughter to collect my medication there. Because of the charges, I decided to request a transfer to a closer facility for treatment

**Attitude of healthcare workers and confidentiality**   The most significant negative determinants for the young persons and key population were confidentiality and poor attitude of healthcare workers. These participants expressed that their negative experiences stemmed from negative attitudes of healthcare workers, including stigmatizing behavior characterized by unfriendly demeanor, lack of empathy, and disrespectful behavior by these healthcare workers. As one of the respondents put it

*"Yes. They are harsh. Sometimes, they will be harsh to you, and I will be like, I was not the one that put myself in this situation, so you can't be harsh to me because of my condition because life goes around. You do not know where you will meet me, and I will help."*

To further buttress this point, another participant stated that *"sometimes they start shouting at clients, not realizing that these clients need care. If they shout at me, I won't come back. Some people, like me, might avoid taking their medications and that's not good"*

Another factor that negatively influenced participants' experience at the DSD service delivery points was the fear of accidental disclosure. This results when service providers knowingly or unknowingly disclose recipients of care status to others without their consent. An example was given by one of the participants who said

*"I know someone who works as a medical records staff at the Federal Medical Center. One day, I found myself in the same community where he lives. He called me, we greeted each other, and I left. After I left, he told someone, 'Do you see that guy? He collects drugs at FMC.' The other person came and mentioned it to me. "*

A young participant referring to the fear of accidental disclosure said that " *What I did not like was that I knew some of the individuals working at the pharmacy. I thought it would just be the head of the pharmacy, but his children were in the pharmacy as well, and some of the staff were not health workers; they were just regular employees. I felt I will be stigmatized and my privacy will not be secured."*

Yet another participant mentioned that *"The only thing I don't like about it is that in my area, some people know the healthcare providers, including those who deliver the drugs. They know the person who brings my drugs, his role in the ART unit, and his status. So there is this stigma attached to you if you associate yourself with him. This is the only problem I have with home delivery or home refill"*

**Theme 3: Suggestions on how to improve DSD services.** In exploring suggestions to enhance Differentiated Service Delivery (DSD) services for HIV care, participants provided valuable insights into various strategies to improve service quality and accessibility. The themes highlighted encompass a wide range of recommendations, each addressing specific areas of improvement within the healthcare system. Notably, respondents underscored the importance of enhancing healthcare worker training to ensure proficiency in delivering DSD services (Training of HCW), aligning healthcare practices with the specific needs of people living with HIV (Engaging HCW in tune with the needs of PLHIV), and increasing awareness about DSD among PLHIV and healthcare workers.

**Training of healthcare workers (HCW)**   Respondents underscored the critical need for comprehensive and ongoing training programs tailored to healthcare workers to ensure their proficiency in delivering Differentiated Service Delivery (DSD) services. They emphasized the importance of equipping HCWs with the necessary skills, knowledge, and competencies to effectively address the unique needs of people living with HIV (PLHIV) within the DSD framework.

**Engagement of healthcare workers (HCW) who are PLHIV**   Engaging healthcare workers (HCWs) who deeply understand the needs of the recipients of care is crucial for fostering empathy, understanding, and peer support in facility and community service delivery points. This is especially important for the key population and young persons. HCWs with lived experience of HIV bring a unique perspective that enhances patient-provider relationships, reduces stigma, and promotes inclusivity. Their visibility as role models and advocates helps challenge misconceptions and discrimination, while peer support initiatives contribute to improved treatment outcomes and patient empowerment. Overall, the engagement of HCWs who are PLHIV plays a vital role in advancing patient-centered HIV care and promoting a supportive healthcare environment.

**DSD awareness programs**   Individuals stressed the significance of implementing robust and targeted awareness programs to enhance PLHIVs' understanding and uptake of DSD services. They emphasized the need for comprehensive educational campaigns aimed at PLHIVs and healthcare providers to promote awareness, dispel myths, and foster informed decision-making regarding DSD participation.

## Discussion

In this study, we identified barriers and enablers to accessing DSD services by PLHIV and matched this with healthcare workers' perceptions regarding DSD and clients' preferences. The study showed that both healthcare workers and PLHIV viewed DSD as an acceptable treatment option that can lead to positive outcomes and that convenience, cost considerations, and healthcare worker attitudes are key factors that influence PLHIV experiences with DSD and can affect the decision to be devolved and retained in DSD.

Convenience of the PLHIV emerged as a key consideration for healthcare providers and clients in choosing DSD models. This correlates with findings from studies in Nigeria [21] and Ghana [22]. The convenience of having drugs delivered to their homes or picked up at their preferred time from a pharmacy was a significant consideration that influenced a positive experience for the PLHIV in these studies. Recipients of care in South Africa also reported a preference for Home delivery of medication, as it relieves them from spending a lot of time waiting in hospital queues [23]. While the Nigerian [21] and Ghana[22] studies focused on a narrower demographic of PLHIV, our study explored the perception of a wider PLHIV group. While the study from Ghana [22] focused on care recipients from a single facility, the Nigerian study [20] focused on persons devolved to DSD in Northern Nigeria. We explored the experience of persons devolved across other regions of Nigeria, including the North. This enables us to explore the social and demographic characteristics that can affect devolvement on a larger scale.

Cost had both a positive and negative effect on clients experience on DSD. While reducing the transport cost positively influenced the DSD experience, payment at the DSD service points resulted in a significant negative experience for PLHIV. Cost of service delivery was also reported as a concern by recipients accessing DSD in South Africa [15]. It is pertinent to note that while cost was identified to possibly play a role in discouraging PLHIV from accessing DSD services in certain instances, it also had a positive influence in their decision to accept services in these service points. Studies have identified DSD as a cost-saving alternative to traditional in-hospital service delivery models, especially in regions of sub-Saharan Africa where there is pervasive poverty [23]. The cost savings in these studies were more pronounced in models that incorporated 6-monthly visits. This could be from the reduced cost of transport to service delivery points. Another study also identified reduction in the cost of visit as a factor that increased acceptance of DSD by recipients of care [24]. From our analysis, we identified cost as a greater concern to women. This is similar to findings from a study in Uganda among female sex workers assessing DSD, where the cost of transport to health facilities was also identified as a barrier to the uptake of HIV services from the healthcare facilities and positively influenced their acceptance of DSD [25]. While this study focused exclusively on one of the priority populations for HIV; Female sex workers, our study explores the attitudes of Female PLHIV in the general population. These populations' needs and experiences vary, which can influence their DSD experience. The greater concern of women with cost could be linked to the fact that women in these populations often have less income and fewer opportunities to make money. A Central Bank of Nigeria report in 2019 identified that the gender financial inclusion gap between males and females is widening. It attributed this to factors such as lower income and education levels in women. The study also attributed the reduced financial inclusion for women to other religious and cultural barriers [26]. Another report also identified that women have less employment and own less businesses compared to males in Nigeria. This is despite the presence of several regulations that promote financial inclusion for women and the country having an almost equal number of males and females in the population [27].

This study also identified concerns around confidentiality and the risk of stigma as additional factors influencing participant experience with DSD. The risk of accidental disclosure was a barrier to accessing care in community-based DSD models. This risk was heightened by poor healthcare worker attitudes during community drug delivery. The studies in Nigeria [20] and Uganda [25] referenced above also identified stigma as a barrier to accessing services in the community. Similar to the findings in the Ghana study [22], inadvertent disclosure during out-of-facility service delivery was one of the major sources of stigma to the PLHIV. Educational activities targeted at healthcare workers, recipients of care, and the general population may be of help in addressing these issues regarding stigma [28,29].

The study identified the need to continuously build healthcare worker capacity on client engagement as one recommendation for improving services at the DSD service delivery points. This training will improve healthcare workers' ability to deliver quality services and build their capacity in non-discriminatory service provision. This may involve training healthcare workers on cognitive behavioral therapy and providing mental health and psychosocial support services.

At the program planning level, additional efforts should be made to advocate for removing or subsidizing service charges at DSD service delivery points. This will increase the acceptance of these services and reduce the financial burden on persons accessing DSD services. While a case could be made for leaving the service charge at these DSD sites, further analysis needs to be done to determine the optimal amount to be paid at these sites and the modalities for payment. If it is determined that these payments are essential, a payment option could be to incorporate these costs as part of routine healthcare costs covered by national health insurance; this will improve accessibility and further streamline the payment processes.

Feedback from the KII with service providers identified that client convenience and considerations for reducing health worker workload topped the reasons for devolving PLHIV to DSD models, while the need to address transport cost received the least consideration. This finding is similar to results from Nigeria (21) and South Africa [30] where factors such as reduction in healthcare worker workload, minimizing queues and increasing client convenience were perceived as key benefits of DSD by the healthcare providers. These were similar to findings from a scoping review [24]. While it is

encouraging that healthcare providers also considered client needs and convenience in their devolvement decisions, it will be encouraging to see other considerations such as client distance to the facility and transport costs receive increased consideration during devolvement decisions. This recommendation is based on findings from the focus group discussions (FGD), where participants highlighted reducing transportation costs as one of the primary factor influencing their choice to transition to a differentiated service delivery (DSD) model. This priority is further emphasized when the socioeconomic situation of clients are put in perspective.

## Conclusion

The findings of this study highlight the multifaceted factors influencing individuals' experiences and perceptions of DSD models. While there is considerable awareness and acceptance of DSD services among participants, several challenges and areas for improvement exist. These challenges as perceived PLHIV include but not stigma, cost of care, and attitude of Healthcare workers.

Addressing these challenges will require a concerted effort from healthcare providers, policymakers, and PLHIV. Strategies to enhance healthcare worker training, optimize service delivery points to promote confidentiality, reduce financial barriers, and streamline service delivery PLHIV are crucial for optimizing DSD models and ensuring equitable access to HIV care and treatment. Furthermore, prioritizing patient-centered approaches and engaging with community-based organizations and peer networks will be essential for reaching marginalized populations and addressing gaps in HIV education and awareness. HIV program managers need to be cognizant of the role cost of services plays in the decision of PLHIV to deploy. Considerations while establishing DSD models should include the affordability of services and cost of travel to access these services. By addressing these challenges and implementing targeted interventions addressing confidentiality, accessibility, and affordability, Nigeria can strengthen its HIV service delivery system and improve health outcomes for individuals living with HIV across Nigeria.

### Limitations and future considerations

While the study identified several barriers to DSD uptake, it did not fully explore how systemic issues like funding limitations, policy constraints, or the impact of broader societal attitudes on service delivery and patient experiences and how these influence the uptake of DSD. Furthermore, the study did not allow for a nuanced exploration of the experiences of key population, as all the typologies were lumped up in one FGD group. In addition to a deeper exploration of the experiences of the key population typologies with regards differentiated services, Further studies examining the impact of broader issues, such as funding limitations and societal perception, on patient experience and DSD uptake are recommended. These studies can also explore how these factors affect service providers. Also, a more in-depth exploration of the different DSD models in Nigeria is needed. These studies should examine how societal influences and administrative concerns affect service delivery at these service points. This exploration should include an assessment of the service quality at these sites vis-a-vis the traditional treatment sites, especially regarding workload distribution, stigma risk, and service delivery quality. Additionally, a Cost-benefit analysis of the different DSD models in Nigeria should be conducted as part of a wider HIV program evaluation. Results of these studies will inform refinement of DSD programs to increase their efficiency, acceptability and effectiveness.

## Author contributions

**Conceptualization:** Johnbaptist Ezenduka, Peter Nwaokennaya, Gbenga Benjamin Obasa, Geoffrey Ogbeke, Ogonna Onuorah.

**Data curation:** Peter Nwaokennaya, Ogonna Onuorah, Lawal Abubakar, Ginigeme Ogochukwu.

**Formal analysis:** Johnbaptist Ezenduka, Ogonna Onuorah, Lawal Abubakar.

**Investigation:** Geoffrey Ogbeke.

**Project administration:** Johnbaptist Ezenduka, Peter Nwaokennaya, Gbenga Benjamin Obasa, Geoffrey Ogbeke, Adebobola Bashorun, Adewale Akinjeji.

**Supervision:** Adebobola Bashorun, Ginigeme Ogochukwu, Adewale Akinjeji.

**Validation:** Adebobola Bashorun, Adewale Akinjeji.

**Writing – original draft:** Johnbaptist Ezenduka, Gbenga Benjamin Obasa.

**Writing – review & editing:** Johnbaptist Ezenduka, Gbenga Benjamin Obasa, Ginigeme Ogochukwu, Emerenini Franklin, Adewale Akinjeji.

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
