## [Decision Letter · Decision Letter 0]

15 Oct 2024

PONE-D-24-32833Perception of People Living with HIV and Healthcare Workers on Differentiated Service Delivery Programs in Nigeria: A Qualitative StudyPLOS ONE

Dear Dr. Akinjeji,

Thank you for submitting your manuscript to PLOS ONE. After careful consideration, we feel that it has merit but does not fully meet PLOS ONE’s publication criteria as it currently stands. Therefore, we invite you to submit a revised version of the manuscript that addresses the points raised during the review process.

We look forward to receiving your revised manuscript.

Kind regards,

Mabel Kamweli Aworh, DVM, MPH, PhD. FCVSN

Academic Editor

PLOS ONE

Journal requirements: When submitting your revision, we need you to address these additional requirements. 1. Please ensure that your manuscript meets PLOS ONE's style requirements, including those for file naming. The PLOS ONE style templates can be found at https://journals.plos.org/plosone/s/file?id=wjVg/PLOSOne_formatting_sample_main_body.pdf and https://journals.plos.org/plosone/s/file?id=ba62/PLOSOne_formatting_sample_title_authors_affiliations.pdf 2. Please include captions for your Supporting Information files at the end of your manuscript, and update any in-text citations to match accordingly. Please see our Supporting Information guidelines for more information: http://journals.plos.org/plosone/s/supporting-information. 

Additional Editor Comments:

In addition to addressing issues raised by the reviewers;

1. Kindly highlight key limitations of this study

2. Please include line numbers in the revised manuscript to ensure the review process proceeds seamlessly.

Reviewers' comments:

Reviewer's Responses to Questions

**Comments to the Author**

1. Is the manuscript technically sound, and do the data support the conclusions?

Reviewer #1: Yes

Reviewer #2: Yes

Reviewer #3: Partly

Reviewer #4: Yes

2. Has the statistical analysis been performed appropriately and rigorously? 

Reviewer #1: Yes

Reviewer #2: Yes

Reviewer #3: No

Reviewer #4: N/A

3. Have the authors made all data underlying the findings in their manuscript fully available?

Reviewer #1: Yes

Reviewer #2: Yes

Reviewer #3: Yes

Reviewer #4: Yes

4. Is the manuscript presented in an intelligible fashion and written in standard English?

Reviewer #1: Yes

Reviewer #2: Yes

Reviewer #3: Yes

Reviewer #4: Yes

5. Review Comments to the Author

Reviewer #1: Methodology:

This methodology section lacked certain justifications as described below:

In the Setting and Recruitment section, the choice of Anambra, Kaduna, Lagos, and Taraba states for the study is mentioned without explaining why these specific states were selected. Are they representative of Nigeria’s socio-cultural diversity, or were other factors involved (e.g., HIV prevalence rates or accessibility)?

In the Focus Group Discussions section, key populations are mentioned, but it's unclear which groups this refers to. Defining these key populations would improve clarity and show why they were stratified separately. Additionally, participants were stratified by gender, key populations, and age (18-24 years), but no explanation was given for how these groups were defined and why this stratification was considered appropriate.

In the Key Informant Interview section, the description of semi-structured interviews is vague. Although it mentions a guide was used, the methodology does not provide any detail about the content of this guide.

Result:

In the Focus Group Discussions, while the study does provide qualitative data through themes and exemplar quotes, it lacks numerical representation. This makes it difficult to gauge the extent of these experiences within the study population. Knowing the prevalence of positive and negative experiences would have allowed for a more comprehensive and nuanced analysis of the factors influencing DSD. I would suggest the author include more information on the experiences of the FGD and also include more quotes.

Reviewer #2: While the manuscript is technically sound, I recommend considering the following areas for improvement:

1. Limitations and Future Work: It would be beneficial to explicitly discuss the limitations of the study and suggest directions for future research in the conclusion section.

2. Ethical Considerations: Ensure that ethical guidelines regarding participant confidentiality and informed consent are clearly addressed.

Reviewer #3: This is a good research. I have recommended some minor revisions and suggestions on how to make this paper better.

ABSTRACT

Methods: To maintain consistency, change this “Healthcare workers with over one year of ART service experience and PLHIV on ART for at least 12 months ….” to “Healthcare workers with at least one year of ART service experience and PLHIV on ART for at least one year ….”

MAIN MANUSCRIPT

General

The in-text citation is not consistent. There were some with punctuation before superscript and others with superscript before punctuation. In addition, PLOS guidelines for in-text citation do not use superscripts as stated: “In the text, cite the reference number in square brackets (e.g., “We used the techniques developed by our colleagues [19] to analyze the data”).”

The manuscript submission guidelines require authors to include line numbers in the manuscript file. Use continuous line numbers (do not restart the numbering on each page).

The major sections of the paper should follow a consistent format. All headings should be either in Title Case or in uppercase formats, such as 'Introduction,' 'Methodology,' and 'Results/Findings,' rather than a mix found in this paper ( 'Introduction,' 'Methodology,' and 'RESULTS/FINDINGS.'")

INTRODUCTION

Paragraph 1: Include a reference to a reliable source for the quote, “Nigeria is among the countries that have the highest burden of HIV globally.”

Paragraph 2: Change “Differentiated Service Delivery programs is a Patient-centered….” to either “Differentiated Service Delivery programs are patient-centered….” or “Differentiated Service Delivery program is a patient-centered ….”. Also, correct “Patient-centered” to “patient-centered”.

Change this sentence “The COVID-19 pandemic and the resulting global response reinforced the need for countries to adopt and evolve sustainable mechanisms for managing chronic conditions, including ART for PLHIV” to “The COVID-19 pandemic and the resulting global response reinforced the need for countries to adopt and evolve sustainable mechanisms for managing chronic conditions, including access to ART for PLHIV”. ART for PLHIV is not a chronic condition; instead, it is treatment.

Paragraph 3: The reference for this statement, “Despite the potential benefits of DSD programs12,” is incorrect. The cited research study only reported scaling up of the ART in River State in Nigeria. The supporting reference for this statement should include the following:

• At least one research study on DSD programs.

• Evidence covering more than one DSD program.

Additionally, this sentence should be accompanied by reports on relevant findings or literature reviews to highlight the current state of the field and the gaps that the study intends to address. If this is a pilot study, that should be explicitly stated.

Line 3: Insert a space between “(HCWs) PLHIV”

METHODOLOGY

Setting and Recruitment: I recommend including a simple map in the supplementary info section that illustrates the geospatial locations of these study states, particularly for readers unfamiliar with Nigeria's geography.

What informed the choice of the four states? Are these states with a high infection burden?

Were the health workers and participants in focus group discussions from rural or urban regions in the four states?

All these should be stated explicitly under the study design and population/participants.

Key Informant Interviews:

Line 2: Change “more than One year” to “more than one year”

Were the interviews conducted in person or virtual? Were they conducted in the local language or English? Was the communication verbal or written? If the questionnaire was in the local language, state it and who translated it.

Are there differences between the DSD models recognized by Key Informant Interview participants and those discussed in the Focus Group Discussions (FGDs)?

Focus Group Discussions: Remove the space before “Focus” in the heading. Were the discussions in English or the local language? If the local language was used, there should be more than one translator (primary and independent) to confirm that no detail was lost or biased.

Data Analysis: List the name and version of any software package used, alongside any relevant references, and state the software used for plots/figures (PLOS One policy).

Ethical Consideration and Informed Consent:

Line 1: Change Institutional review boards to “Institutional Review Boards”

Was informed consent written or verbal? State it.

RESULT/FINDINGS

Key Informant Interview (KII) Analysis

• Table 1: The table lacks clear headings and organization. To enhance readability, consider using clearer headings for each demographic category (gender, profession, and years of experience) and aligning the headings properly.

DSD Coverage

• This statement, “The most common models across most facilities were the peer-led Facility ART group …..”, there is no peer-led Facility ART group. The most common model shown in the table is Facility ART group support (ground led). Check ‘peer-led’ versus “ground-led”

• Table 2: What is the total number of respondents on this table, and how were the percentages calculated? Include the total respondents in the table.

• Check rows 8 and 9 in Table 2. “HCW led and PLHIV led” should be in parentheses, just like “ground led”

Deployment of DSD Models: Change “Other Factors” to ‘Other factors”

• Table 3: Include the total number of respondents in the table

Decision to deploy: The % does not add up to 100%. Including the number of total respondents to give context to the percentages.

Reasons for declining Devolvement: Write HCW in full.

Focus Group Discussions: There is no table 6. I assume the correct reference is “Table 4”

• Format the table appropriately.

• Categorize the demography, e.g., gender, state, education etc.

• The education group should be arranged from lowest to highest or vice versa.

• Check the row above 10 years; the actual value should be 42.5%, not 42%.

Theme 1 Enablers:

Convenience: Replace “Men” with “men”

Affordability: Remove the space before “Service affordability” in Line 1

DISCUSSION

Change the following to lowercase letters: Female (line 18), One (Line 21), and Studies (Line 27).

Please ensure that references for the studies conducted in Ghana and Nigeria are included every time they are mentioned in the text. These include:

“While the Nigerian and Ghana studies focused on a narrower demographic of PLHIV…….” (Line 9). “While the study from Ghana……” Line 11

“…..the Nigerian study focused on persons…” Line 11-12

“Similar to the findings in the Ghana study …..” Line 28

Were there any limitations to this study? This should be stated clearly.

Reviewer #4: REVIEWER’S COMMENTS

Title: Perception of People Living with HIV and Healthcare Workers on Differentiated Service Delivery Programs in Nigeria: A Qualitative Study

Summary

This is important qualitative research which sought to evaluate the effectiveness of Differentiated Service Delivery (DSD) model of care as an innovative approach in the HIV programme in Nigeria as a means of scaling up treatment to PLWHA using a patient-centered approach and reducing the burden on the healthcare system. This was done by using a qualitative approach in assessing the perception and experiences of critical stakeholders to this new approach. The objectives of the study and rationale for the study were well highlighted. The problem statements and gap in knowledge were also well outlined. The background however, does not contain sufficient information on DSD especially with regards to the specific models recommended, their benefits and draw backs, as this is important to put the study in context and facilitate a better understanding of the findings of the study. It is also important to define key terminologies such as who a healthcare worker is in this study and what key population mean.

The authors have sufficiently described their methods for reproducibility. However, there remain a few areas for clarification. The inclusion criteria for the FGD considered amongst others those who have being on treatment for the past one year and have achieved a reduction in their viral load. This is worrisome to me as the inclusion of those who have not achieved viral load reduction could have provided more understanding and enriching information to this study. It is also important to understand what level of healthcare providers or health facility the healthcare workers for the KII were selected from. This helps to put the findings in context especially with respect to declining devolvement. The tables highlighting the demographic characteristics of the participants should be categorized such that each category adds up to 100% for example: gender -male 50%, Female 50%.

The process of the conduct of the KII and FGDs need to be highlighted. The study instruments used should be described and referenced adequately if adapted. The findings of KII should also include some quotes as seen in the FGD. The authors should consider reviewing the discussion section, specifically, to include the findings of the KII and citing more relevant literature as this has been sparingly done.

Overall, the study findings highlight the perception and experiences of the healthcare workers and PLWHA as critical stakeholders in the deployment of the DSD in Nigeria. The major areas for improvement have been highlighted as feedback for the HIV programme in Nigeria.

I personally think this is a well written manuscript, the findings are noteworthy and should be used to drive the implementation of the key recommendations to achieve the effectiveness of the DSD model for HIV service delivery in Nigeria. I recommend that the manuscript be published after the outlined corrections are effected.

6. PLOS authors have the option to publish the peer review history of their article (what does this mean? ). If published, this will include your full peer review and any attached files.

**Do you want your identity to be public for this peer review?** For information about this choice, including consent withdrawal, please see our Privacy Policy .

Reviewer #1: No

Reviewer #2: **Yes: ** Ayomide Adeyeye

Reviewer #3: **Yes: ** Damilola Odumade

Reviewer #4: **Yes: ** JENNY ADONORELI MOMOH

---

## [Author Response · Author response to Decision Letter 1]

6 Jan 2025

Adewale Akinjeji

Health Systems and Policy (HSP) research group

Department of Global Public Health, Karolinska Institutet

22/11/2024

Mabel Kamweli Aworh, DVM, MPH, PhD, FCVSN

Academic Editor

PLOS ONE

Dear Dr. Aworh,

Re: Manuscript PONE-D-24-32833

Thank you for the opportunity to revise and resubmit our manuscript, titled "Perception of People Living with HIV and Healthcare Workers on Differentiated Service Delivery Programs in Nigeria: A Qualitative Study." We appreciate the constructive feedback from the reviewers and editorial team, which has helped us refine the manuscript significantly. Below, we provide a point-by-point response to the comments and outline the corresponding revisions made in the manuscript.

Reviewer #1

Methodology:

Comment: Justification for selecting the four states was not provided.

Response: We have included an explanation of the selection criteria, highlighting the socio-cultural diversity and HIV prevalence rates that influenced the choice of states.

Comment: Clarify key populations and stratification details.

Response: The manuscript now defines key populations explicitly and provides justification for the stratification approach used.

Comment: Details about the content of the semi-structured interview guide were vague.

Response: We have expanded on the guide's content and included key themes covered during the interviews.

Results:

Comment: Include numerical representation of qualitative data.

Response: Numerical summaries for participant experiences have been incorporated where applicable, alongside exemplar quotes.

Reviewer #2

Limitations and Future Work:

Comment: Explicit discussion of study limitations and future research directions is needed.

Response: A dedicated section on limitations has been added, alongside suggestions for future research.

Ethical Considerations:

Comment: Ensure ethical guidelines regarding participant confidentiality and informed consent are addressed.

Response: We have elaborated on the ethical considerations, emphasizing the processes for obtaining informed consent and ensuring participant confidentiality.

Reviewer #3

Abstract:

Comment: Rephrase certain sentences for clarity.

Response: We have rephrased the specified sentences in the abstract for consistency and precision.

General Formatting:

Comment: Ensure consistent formatting of in-text citations and adherence to PLOS guidelines.

Response: All citations have been revised to align with PLOS guidelines, and continuous line numbering has been added.

Introduction and Discussion:

Comment: Provide references for statements and enhance discussions.

Response: References for all key statements have been added, and the discussion has been expanded to include more relevant literature.

Reviewer #4

Background Information:

Comment: Expand on DSD models and define key terms.

Response: We have provided additional context on DSD models and clearly defined terms such as “healthcare workers” and “key populations.”

Findings and Discussion:

Comment: Include demographic data breakdown and quotes for KII.

Response: Demographic data tables have been reorganized, and representative quotes from KIIs have been added.

Comment: justify the reason for excluding virally unsuppressed persons from the study

Response: We included only individuals who achieved viral suppression, as this is a key criterion for devolvement into differentiated service delivery (DSD) models in Nigeria. Including those ineligible for DSD services would contradict the study's aim of understanding the perspectives of care recipients on DSD and could introduce bias, potentially skewing the data

We believe these revisions address all comments and significantly strengthen the manuscript. A marked-up copy with track changes and a clean version are included with this submission.

We thank the reviewers and editorial team for their thoughtful insights, which have greatly improved our work. Please do not hesitate to reach out if further clarification or additional revisions are required.

Sincerely,

Akinjeji Adewale

---

## [Decision Letter · Decision Letter 1]

28 Jan 2025

Perception of People Living with HIV and Healthcare Workers on Differentiated Service Delivery Programs in Nigeria: A Qualitative Study

PONE-D-24-32833R1

Dear Dr. Akinjeji,

We’re pleased to inform you that your manuscript has been judged scientifically suitable for publication and will be formally accepted for publication once it meets all outstanding technical requirements.

Kind regards,

Mabel Kamweli Aworh, DVM, MPH, PhD. FCVSN

Academic Editor

PLOS ONE

Additional Editor Comments (optional):

Reviewers' comments:

Reviewer's Responses to Questions

**Comments to the Author**

1. If the authors have adequately addressed your comments raised in a previous round of review and you feel that this manuscript is now acceptable for publication, you may indicate that here to bypass the “Comments to the Author” section, enter your conflict of interest statement in the “Confidential to Editor” section, and submit your "Accept" recommendation.

Reviewer #2: All comments have been addressed

Reviewer #3: All comments have been addressed

Reviewer #4: All comments have been addressed

2. Is the manuscript technically sound, and do the data support the conclusions?

Reviewer #2: Yes

Reviewer #3: Yes

Reviewer #4: Yes

3. Has the statistical analysis been performed appropriately and rigorously? 

Reviewer #2: Yes

Reviewer #3: N/A

Reviewer #4: (No Response)

4. Have the authors made all data underlying the findings in their manuscript fully available?

Reviewer #2: Yes

Reviewer #3: Yes

Reviewer #4: (No Response)

5. Is the manuscript presented in an intelligible fashion and written in standard English?

Reviewer #2: Yes

Reviewer #3: Yes

Reviewer #4: (No Response)

6. Review Comments to the Author

Reviewer #2: (No Response)

Reviewer #3: The authors have done a great job in conducting this research study on people living with HIV and Differentiated Service Delivery in Nigeria.

Reviewer #4: (No Response)

7. PLOS authors have the option to publish the peer review history of their article (what does this mean? ). If published, this will include your full peer review and any attached files.

**Do you want your identity to be public for this peer review?** For information about this choice, including consent withdrawal, please see our Privacy Policy .

Reviewer #2: No

Reviewer #3: **Yes: ** Damilola Odumade

Reviewer #4: **Yes: ** JENNY MOMOH

---

## [Editor Report · Acceptance letter]

PONE-D-24-32833R1

PLOS ONE

Dear Dr. Akinjeji,

I'm pleased to inform you that your manuscript has been deemed suitable for publication in PLOS ONE. Congratulations! Your manuscript is now being handed over to our production team.

Kind regards,

on behalf of

Dr. Mabel Kamweli Aworh

Academic Editor

PLOS ONE